# The Influence of High Blood Pressure on Developing Symptomatic Lumbar Epidural Hematoma after Posterior Lumbar Spinal Fusion Surgery: Clinical Data Warehouse Analysis

**DOI:** 10.3390/jcm11154522

**Published:** 2022-08-03

**Authors:** Jin-Seo Yang, Young-Suk Kwon, Jong-Ho Kim, Jae-Jun Lee, Eun-Min Seo

**Affiliations:** 1Department of Neurosurgery, Chunchon Sacred Heart Hospital, Hallym University College of Medicine, Chuncheon 24252, Korea; yang@hallym.or.kr; 2Division of Big Data and Artificial Intelligence, Institute of New Frontier Research, Chuncheon Sacred Heart Hospital, Hallym University College of Medicine, Chuncheon 24253, Korea; gettys@hallym.or.kr (Y.-S.K.); poik99@hallym.or.kr (J.-H.K.); iloveu59@hallym.or.kr (J.-J.L.); 3Department of Anesthesiology and Pain Medicine, College of Medicine, Chuncheon Sacred Heart Hospital, Hallym University College of Medicine, Chuncheon 24253, Korea; 4Department of Orthopedic Surgery, Chuncheon Sacred Heart Hospital, Hallym University College of Medicine, Chuncheon 24253, Korea

**Keywords:** symptomatic lumbar epidural hematoma (SLEH), posterior lumbar spinal fusion surgery, clinical data warehouse (CDW), systolic blood pressure (SBP)

## Abstract

Background: Determining the risk factors for symptomatic lumbar epidural hematoma (SLEH) is important for preventing postoperative SLEH. However, the relationship between blood pressure and SLEH is still debatable. The purpose of our study was to determine the risk factors for postoperative SLEH, to assess the influence of high blood pressure on developing SLEH after posterior lumbar spinal fusion surgery, and to evaluate the usefulness of big data analysis utilizing a clinical data warehouse (CDW). Methods: The clinical data of patients who had undergone posterior lumbar spinal fusion surgery were acquired from the CDW of Hallym University Medical Center. The acquired clinical data were compared between patients without postoperative SLEH and with postoperative SLEH. Results: Postoperative SLEH that required hematoma evacuation surgery within 72 h after posterior lumbar spinal fusion surgery occurred in 17 (1.3%) of 1313 patients. According to the multivariate logistic regression analysis, the risk factors for postoperative SLEH are platelet count difference (OR 1.28, *p* = 0.03), postoperative international normalized ratio (INR) difference (OR 31.4, *p* = 0.028), and postoperative systolic blood pressure (SBP) difference (≥10 mmHg) (OR 1.68, *p* = 0.048). An increase in postoperative SBP (OR 1.68, *p* = 0.048) had a statistically significant influence on the occurrence of postoperative SLEH. Conclusions: Big data analysis utilizing a CDW could be useful for extending our knowledge of the risk factors for postoperative SLEH and preventing postoperative SLEH after posterior lumbar spinal fusion surgery.

## 1. Introduction

Symptomatic lumbar epidural hematoma (SLEH) is a rare but serious complication after spinal surgery. SLEH has a typical onset time of several hours to days after spinal surgery. Although magnetic resonance imaging (MRI) revealed that asymptomatic lumbar epidural hematoma is experienced by 15–89% of patients after spinal surgery [1,2,3], SLEH was shown to occur in 0.1–0.7% of patients [1,2,3]. The symptoms of lumbar epidural hematoma are severe back and leg pain, motor weakness, sensory changes, and bladder and bowel dysfunction. In most cases, postoperative SLEH requires another surgery for hematoma evacuation and decompression to restore neurologic function. When the diagnosis and treatment of postoperative SLEH are delayed, chronic intractable leg pain and permanent paralysis sometimes remain [4]. Thus, the determination of the risk factors for postoperative SLEH is important for preventing postoperative SLEH. According to previous studies, the risk factors for postoperative SLEH include old age (>60 years), alcoholism, NSAIDs, coagulopathy, high estimated blood loss, postoperative drain output, multilevel procedures, and previous spinal surgery.

Some risk factors for postoperative SLEH remain debatable because of the inconsistent outcomes of previous studies. In some studies, an increase in systolic blood pressure (SBP) (>50 mmHg) after extubation was found to be a risk factor for postoperative SLEH, but preoperative hypertension or preoperative medication for hypertension was not [5]. However, other studies reported inconsistent outcomes (preoperative hypertension was found to be a risk factor for postoperative SLEH) [6,7,8]. The influence of high blood pressure on the incidence of postoperative SLEH remains unclear in patients undergoing spinal surgery.

There are numerous risk factors for postoperative SLEH. Thus, it is time-consuming to evaluate all possible factors (i.e., demographic profile, laboratory data, perioperative factors, etc.) and to determine all the risk factors. Nowadays, clinical data are rapidly digitized, and big data analysis facilitates the management of clinical data for medical practice and research [9].

Thus, the purpose of our study was to determine the risk factors for postoperative SLEH, to assess the influence of high blood pressure on developing SLEH after posterior lumbar spinal fusion surgery, and to evaluate the usefulness of big data analysis utilizing a clinical data warehouse (CDW).

## 2. Materials and Methods

### 2.1. Ethics and Data Source

This study was performed retrospectively with the approval of the clinical research deliberation committee of Hallym University Chuncheon Sacred Heart Hospital. The study included vulnerable patients but was exempted from the patient consent requirement because data were collected after all treatment processes were completed. All data were acquired from the clinical data warehouse of Hallym University Medical Center. The CDW contained data from five hospitals in Hallym University. In addition to electronic medical records, the CDW included formal and unstructured data such as prescriptions, laboratory test results, and imaging test results.

### 2.2. Participants

Participants in this study were patients who had undergone only posterior lumbar spinal fusion surgery, including posterolateral fusion or posterior lumbar interbody fusion, from January 2012 to February 2022. The exclusion criteria were patients under the age of 18, patients who had undergone other surgeries at the same time, and patients with data deficits.

### 2.3. Exposure Variables and Primary Outcomes

The exposure variable in this study was the difference between postoperative SBP and high normal blood pressure (BP). The high normal SBP was 120 mmHg. If the postoperative SBP was less than 120 mmHg, there was no difference. Postoperative SBP was measured three times (1 h, 2 h, and 3 h post operation) in the postoperative ward, and the average was calculated. The primary result was postoperative SLEH that required hematoma evacuation surgery within 72 h after posterior lumbar fusion surgery.

### 2.4. Other Variables

Other covariates included old age (≥70 years old), male gender, obesity (BMI > 29.9), history of hypertension and diabetes, smoking, American Society of Anesthesiologists physical status >2, preoperative nonsteroidal anti-inflammatory drugs, anti-platelet drugs and anti-coagulants, difference from baseline in pre- and postoperative platelet count (baseline: 150,000/uL), international normalized ratio (INR) (baseline: 1.0) and partial thromboplastin time (aPTT) (baseline: 35 s), spinal disease, surgical level range, intraoperative time, and use of anti-hypertensive drugs. Postoperative blood tests were taken 3 h post operation.

### 2.5. Statistics

We expressed sequential data as median and interquartile ranges (IQRs), and categorical data as frequencies and percentages. Sequential variables were evaluated by *t*-test or Mann–Whitney test, and categorical variables were evaluated by chi-square analysis. The unadjusted and adjusted odds ratio and 95% confidence intervals for the occurrence of SLEH were evaluated by using logistic regression. *p*-value < 0.05 was regarded as statistically significant. Statistical analyses were conducted using IBM SPSS Statistics (version 26.0; IBM Corp., Armonk, NY, USA).

## 3. Results

From January 2012 to 3 February 2022, 1708 people underwent posterior lumbar spinal fusion surgery, including posterolateral fusion or posterior lumbar interbody fusion. Of these, 18 people were under the age of 18, 57 had undergone other operations at the same time, and 320 had data deficits; thus, 1313 people were finally included in the study. Postoperative SLEH that required hematoma evacuation surgery within 72 h after posterior lumbar spinal fusion surgery occurred in a total of 17 (1.3%) patients. Multilevel operations (surgical level > 2) had a prevalence of 11.8% (*n* = 2) in the SLEH group and 24.9% (*n* = 323) in the control group (no SLEH); however, there was no statistically significant difference (*p* = 0.212). The operation time (hours) was 3.1 (1.9 to 4.4) in the SLEH group and 3.4 (2.4 to 4.5) in the control group (no SLEH), but there was no significant difference between the groups (*p* = 0.378). The patients’ demographic and perioperative data are summarized in Table 1.

According to the results of the univariate analysis, anti-platelet drugs, pre- and postoperative platelet count difference, pre- and postoperative INR difference, and blood loss > 2000 mL were risk factors for postoperative SLEH. Among them, there were significant differences in postoperative platelet count difference (OR 1.28, 95% confidence interval, 1.02–1.59; *p* = 0.03) and postoperative INR difference (OR 31.4, 95% confidence interval, 1.46–676.41; *p* = 0.028).

Patients who had preoperative hypertension accounted for 29.4% (*n* = 5) of the SLEH group and 47.7% (*n* = 618) of the control group (no SLEH), but there was no statistically significant difference (*p* = 0.134). The difference between postoperative systolic blood pressure and high normal blood pressure (120 mmHg) was 7 (0 to 15) in the SLEH group and 0 (0 to 9) in the control group, with no significant difference between the groups. Additionally, there were no significant differences in the other variables (mean age, gender, BMI, operation time, surgical level, or preoperative anti-coagulant usage).

When SBP increased by 10 mmHg from 120 mmH after surgery, the unregulated odds ratio and adjusted odds ratio were 1.65 (95% confidence interval, 1.11–2.45; *p* = 0.014) and 1.68 (95% confidence interval, 1.00–2.82; *p* = 0.048), respectively. Therefore, a ≥ 10 mmHg increase in postoperative SBP had a statistically significant effect on the incidence of postoperative SLEH.

## 4. Discussion

According to previous studies, the occurrence of postoperative SLEH is 0.1–1% [7,10,11].

The occurrence of postoperative SLEH that required hematoma evacuation surgery within 72 h after posterior lumbar spinal fusion surgery was 1.3% in our study. Our result was slightly higher because we included participants who required more aggressive surgery (lumbar spinal fusion surgery) in this study. Although postoperative SLEH is rare following spinal surgery, its occurrence is disastrous, as it causes neural compression, which usually results in neurologic dysfunction [8].

Previous studies have reported numerous risk factors for postoperative SLEH [2,3,5,6,7,8,10,12,13,14]. Advanced age (≥70 years), multilevel surgery, blood loss, BMI, hypertension, alcohol use, NSAIDs, and increased blood pressure were found to be significant risk factors in some papers but not in others [2,5,6,7,8,10,11,14,15]. Additionally, the use of anti-coagulant and/or anti-platelet drugs is considered one of the most important triggers for postoperative SLEH [16,17,18,19].

In several studies, a multilevel surgery was found to be a risk factor for postoperative SLEH [2,6,10]. One reason for this is that the increased surgical duration and exposure can lead to increased bleeding, which contributes to postoperative SLEH [7,10]. Coagulation dysfunction may lead to high estimated blood loss and low postoperative hemoglobin, which is related to a long surgical duration [7]. Furthermore, blood transfusion further aggravates coagulation dysfunction [2], and a long surgical duration increases the chance of epidural venous plexus injury, which may be connected to postoperative SLEH [13]. Thus, to perform hemostasis during surgery is important to reduce the incidence of postoperative SLEH.

According to our univariate analysis, anti-platelet drugs, pre- and postoperative platelet count difference, pre- and postoperative INR difference, and estimated blood loss >2000 mL are risk factors for postoperative SLEH. These factors, like high estimated blood loss, usually lead to coagulation dysfunction, which may be related to postoperative SLEH. Among these factors, a multiple logistic regression analysis demonstrated that postoperative platelet count difference (OR 1.28, *p* = 0.03), postoperative PT INR difference (OR 31.4, *p* = 0.028), and postoperative SBP difference (>10 mmHg) (OR 1.68, *p* = 0.048) were significant essential risk factors. 

There were no significant differences in advanced age (>70 years), BMI, surgical level, anti-platelet drugs, NSAID, or operation time. According to previous studies, postoperative SLEH tends to occur in patients with cerebrovascular or cardiovascular disease and may be related to the use of anti-platelet drugs in patients with this disease. On the other hand, anti-coagulant or anti-platelet drugs were not related to the incidence of postoperative SLEH in our study, because anti-coagulant and anti-platelet drugs were withdrawn at least five days before surgery to reduce perioperative bleeding in our hospital.

Most previous studies have reported that preoperative hypertension or hypertension treatment is not a risk factor for postoperative SLEH [3,5,6]. However, some studies have concluded that a high systolic or diastolic blood pressure is a risk factor for postoperative SLEH.

A high diastolic blood pressure (DBP) may result in the high viscosity of whole blood and the formation of blood clots, causing drainage dysfunction and leading to the compression of neural tissue in the epidural space by accumulated blood [20]. Therefore, a high DBP is related to the incidence of postoperative SLEH [8,20,21,22,23,24,25,26].

During surgery, the BP tends to be low because of anesthesia drugs expanding the peripheral vessels. After the termination of anesthesia, the blood pressure increases due to the lack of anesthesia drugs and postoperative pain. This causes re-bleeding and postoperative SLEH at the operation site. Therefore, a high postoperative BP is related to the incidence of postoperative SLEH [5]. In our study, we evaluated the relationship between a change in SBP and postoperative SLEH. The postoperative SBP difference was estimated as the difference between the postoperative systolic blood pressure and high normal blood pressure (120 mmHg). When the systolic blood pressure increased by 10 mmHg from 120 mmH after surgery, the unregulated odds ratio and adjusted odds ratio were 1.65 (95% confidence interval, 1.11–2.45; *p* = 0.014) and 1.68 (95% confidence interval, 1.00–2.82; *p* = 0.048), respectively. Therefore, a >10 mmHg increase in postoperative SBP had a statistically significant effect on the incidence of postoperative SLEH. 

Therefore, postoperative SBP should be carefully monitored. To prevent postoperative SLEH, controlling the postoperative blood pressure at <120 mmHg as much possible, confirming hemostasis, and inserting suction drainage before closing the wound is essential. If the symptoms of lumbar epidural hematoma develope, emergency surgical decompression is required to save the neurological function.

This study had some limitations. The main limitation of our study was that data were retrospectively analyzed. For example, INR is not always the best measure for monitoring the effect of novel anticoagulants. Therefore, the retrospective INR data may not have been accurate. Another limitation was that postoperative lumbar MRI could not be conducted for all patients. Therefore, we could not analyze the amount and location of postoperative SLEH, the severity of neural compression, and asymptomatic lumbar epidural hematoma by MRI imaging. This enabled the certain diagnosis of only postoperative SLEH in the analysis, but the low probability of diagnosis may have impacted the analysis of the relationship among risk factors. Thus, it may be improper to regulate the results further for possible disruption variables.

## 5. Conclusions

In this study, the risk factors for postoperative SLEH after posterior lumbar spinal fusion surgery were assessed utilizing a CDW. The occurrence of postoperative SLEH that required hematoma evacuation surgery within 72 h after posterior lumbar spinal fusion surgery was 1.3%. When the systolic blood pressure increased by 10 mmHg from 120 mmH after posterior lumbar spinal fusion surgery, the occurrence of postoperative SLEH increased. There was a significant relationship between an increase in postoperative SBP and the occurrence of postoperative SLEH after posterior lumbar spinal fusion surgery. Big data analysis utilizing a clinical data warehouse (CDW) could be useful for extending our knowledge of the risk factors for postoperative SLEH and preventing postoperative SLEH after posterior lumbar spinal fusion surgery.

## Figures and Tables

**Table 1 jcm-11-04522-t001:** Demographic and perioperative data of patients who underwent posterior lumbar spinal fusion surgery.

Variable	No Postoperative Symptomatic Lumbar Epidural Hematoma (Number = 1296)	Postoperative Symptomatic Lumbar Epidural Hematoma (Number = 17)	*p* Value
Old age (≥70 years)	496 (38.3)	3 (17.6)	0.082
Male	606 (46.8)	8 (47.1)	0.98
Obesity (BMI > 29.9)	104 (8.0)	1 (5.9)	0.746
Hypertension	618 (47.7)	5 (29.4)	0.134
Diabetes	311 (24.0)	4 (23.5)	0.964
Smoking	186 (14.4)	2 (11.8)	0.762
ASA PS > 2	526 (40.6)	8 (47.1)	0.589
NSAID	454 (35.0)	5 (29.4)	0.629
Anti-platelet	10 (0.8)	1 (5.9)	0.022
Anti-coagulation	20 (1.5)	1 (5.9)	0.157
Preoperative platelet difference (10,000/uL) (median, IQR)	0 (0 to 0)	0 (0 to 0.7)	0.003
Preoperative INR difference (median, IQR)	0 (0 to 0.1)	0.1 (0.0 to 0.2)	<0.001
Preoperative aPTT difference (seconds) (median, IQR)	0 (0 to 1)	0 (0 to 2.4)	0.696
HNP/stenosis	169 (13.0)/401 (30.9)	2 (11.8)/6 (35.3)	0.927
Surgical level > 2	323 (24.9)	2 (11.8)	0.212
Operation time (hours) (median, IQR)	3.4 (2.4 to 4.5)	3.1 (1.9 to 4.4)	0.378
Anti-hypertensive drug during surgery	1 (0 to 2)	0 (0 to 1)	0.203
Estimated blood loss > 2000 mL	110 (8.5)	4 (23.5)	0.029
Patient-controlled analgesia	1226 (94.6)	16 (94.1)	0.931
Postoperative platelet difference (10,000/uL) (median, IQR)	0 (0 to 0.1)	1.4 (0 to 4.2)	<0.001
Postoperative INR difference (median, IQR)	0.1 (0.0 to 0.2)	0.2 (0.1 to 0.5)	0.001
Postoperative aPTT difference (seconds) (median, IQR)	0 (0 to 0.8)	0 (0 to 7.8)	0.128
Postoperative systolic blood pressure difference (median, IQR)	0 (0 to 9)	7 (0 to 15)	0.118

The unadjusted odds ratio and adjusted odds ratio of other covariates are summarized in Table 2 and Table 3, respectively.

**Table 2 jcm-11-04522-t002:** Unadjusted odds ratio for postoperative symptomatic lumbar epidural hematoma.

Variable	Odds Ratio	*p* Value
Old age (≥70 years)	0.35 (0.1 to 1.21)	0.096
Male	1.01 (0.39 to 2.64)	0.98
Obesity (BMI > 29.9)	0.72 (0.09 to 5.46)	0.747
Hypertension	0.46 (0.16 to 1.3)	0.144
Diabetes	0.97 (0.32 to 3.01)	0.964
Smoking	0.8 (0.18 to 3.51)	0.763
ASA PS > 2	1.3 (0.5 to 3.39)	0.59
NSAID	0.77 (0.27 to 2.21)	0.63
Anti-platelet	8.04 (0.97 to 66.56)	0.053
Anti-coagulation	3.99 (0.5 to 31.54)	0.19
Preoperative platelet difference (10,000/uL)	1.41 (1.19 to 1.66)	<0.001
Preoperative INR difference	37.99 (3.93 to 366.86)	0.002
Preoperative aPTT difference (seconds)	0.97 (0.82 to 1.15)	0.725
Herniated nucleus pulposus	0.83 (0.29 to 2.34)	0.723
Spinal stenosis	0.79 (0.16 to 3.96)	0.775
Surgical level > 2	2.49 (0.57 to 10.95)	0.227
Operation time (hour)	0.86 (0.61 to 1.21)	0.397
Anti-hypertensive drug during surgery	0.65 (0.35 to 1.23)	0.187
Estimated blood loss > 2000 mL	0.3 (0.1 to 0.94)	0.039
Patient-controlled analgesia	1.09 (0.14 to 8.37)	0.931
Postoperative platelet difference (10,000/uL)	1.33 (1.15 to 1.53)	<0.001
Postoperative INR difference	61.96 (12.12 to 316.78)	<0.001
Postoperative aPTT difference (seconds)	1.07 (1.02 to 1.12)	0.156
Postoperative systolic blood pressure difference (10 mmHg)	1.65 (1.11 to 2.45)	0.014

**Table 3 jcm-11-04522-t003:** Adjusted odds ratio for postoperative symptomatic lumbar epidural hematoma.

Variable	Odds Ratio	*p* Value
Old age (≥70 years)	4.56 (0.88 to 23.54)	0.07
Male	1.25 (0.39 to 4.04)	0.706
Obesity (BMI > 29.9)	0.67 (0.08 to 5.89)	0.716
Hypertension	1.92 (0.51 to 7.16)	0.334
Diabetes	1.2 (0.3 to 4.74)	0.796
Smoking	2.16 (0.34 to 13.72)	0.414
ASA PS > 2	1.12 (0.28 to 4.47)	0.874
NSAID	1.51 (0.41 to 5.57)	0.536
Anti-platelet	0.05 (0.0 to 0.59)	0.019
Anti-coagulation	0.39 (0.03 to 5.12)	0.473
Preoperative platelet difference (10,000/uL)	1.1 (0.83 to 1.47)	0.502
Preoperative INR difference	6.8 (0.07 to 634.29)	0.408
Preoperative aPTT difference (seconds)	0.79 (0.58 to 1.08)	0.143
Herniated nucleus pulposus	0.41 (0.11 to 1.61)	0.202
Spinal stenosis	0.73 (0.12 to 4.39)	0.734
Surgical level > 2	4.4 (0.73 to 26.47)	0.106
Operation time (hour)	0.76 (0.49 to 1.15)	0.196
Anti-hypertensive drug during surgery	0.75 (0.36 to 1.55)	0.431
Estimated blood loss > 2000 mL	0.83 (0.12 to 5.68)	0.847
Patient-controlled analgesia	1.92 (0.17 to 21.1)	0.595
Postoperative platelet difference (10,000/uL)	1.28 (1.02 to 1.59)	0.03
Postoperative INR difference	31.4 (1.46 to 676.41)	0.028
Postoperative aPTT difference (seconds)	1.02 (0.95 to 1.11)	0.555
Postoperative systolic blood pressure difference (10 mmHg)	1.68 (1.0 to 2.82)	0.048

## Data Availability

All data were obtained from the clinical data warehouse (CDW) of the five hospitals of Hallym University Medical Center.

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
