# Peer review of "The Influence of High Blood Pressure on Developing Symptomatic Lumbar Epidural Hematoma after Posterior Lumbar Spinal Fusion Surgery: Clinical Data Warehouse Analysis"

_jcm, 2022, doi:10.3390/jcm11154522_

Round 1
Reviewer 1 Report
The incidence of PSEH is a little higher we can not assume only that was because they performed aggressive surgeries. Numerous risk factors have been raised in many papers and Hypertension in post-op patients is controversial. This paper considers a large number of patients in five hospitals. One important limitation is a retrospective study does not allow an analysis of MRI in all patients.Is not original but interesting because of the devastating consequences of this complication for this reason the ability to know more allows us to avoid it.
The conclusion is consistent with the arguments and the statistical evidence and addresses the main question posed.
In general, Is a good paper that gives us circumstantial evidence to face some devastating complications in spine surgery.
Author Response
Comments and Suggestions for Authors
The incidence of PSEH is a little higher we can not assume only that was because they performed aggressive surgeries. Numerous risk factors have been raised in many papers and Hypertension in post-op patients is controversial. This paper considers a large number of patients in five hospitals. One important limitation is a retrospective study does not allow an analysis of MRI in all patients.
Is not original but interesting because of the devastating consequences of this complication for this reason the ability to know more allows us to avoid it.
The conclusion is consistent with the arguments and the statistical evidence and addresses the main question posed.
In general, Is a good paper that gives us circumstantial evidence to face some devastating complications in spine surgery.
Answer: Thank for your very kindly advice.

Reviewer 2 Report
The authors described a retrospective study that measured the impact of various risk factors in the development of post-operative epidural hematoma after lumbar spinal fusion surgeries. Epidural hematomas are dreaded complications and identifying risk factors are highly important.
This study has a reasonable sample size and the statistics appear sound. However, I do have a few recommendations:
1) It will be useful to know the type of procedure and number of levels operated, and their impact on epidural hematoma formation
2) Same for the duration of procedure
3) Please clarify when the postoperative SBP was measured (line 86)
4) Please clarify when the postoperative blood tests were taken (line 94)
5) Table 1 is a bit confusing - is it the n number and then the s.d. in brackets? Seems to apply to the top few rows but not for INR, which seems to be average INR and (range)?
6) Please describe the effect of anticoagulant use in line 173 and your institute’s policy in stopping these pre-operatively
Minor points
1) Please add the OR and p-value for postoperative SBP in the abstract (“Increase of postoperative SBP had statistically significant effect on 29 the incidence of symptomatic PSEH…”)
2) Please note that INR is not always the best measure for monitoring the effect of the novel anticoagulants – another limitation
Author Response
Reviewer 2.
Comments and Suggestions for Authors
The authors described a retrospective study that measured the impact of various risk factors in the development of post-operative epidural hematoma after lumbar spinal fusion surgeries. Epidural hematomas are dreaded complications and identifying risk factors are highly important.
This study has a reasonable sample size and the statistics appear sound. However, I do have a few recommendations:
1) It will be useful to know the type of procedure and number of levels operated, and their impact on epidural hematoma formation
Answer: Thank for your very kindly advice. We sincerely revised our study like below.
Participants in this study were patients who underwent lumbar spinal fusion surgery from January 2012 to February 2022. Thus, our study has single type of procedure (lumbar spinal fusion surgery)
There was no significant difference between both groups. Also, there were no significant differences in other variables (mean age, gender, BMI, operation time, surgical level or preoperative anti-coagulant usage).
2) Same for the duration of procedure
Answer: Thank for your very kindly advice. We sincerely revised our study like below.
There was no significant difference between both groups. Also, there were no significant differences in other variables (mean age, gender, BMI, or preoperative anti-coagulant usage).
3) Please clarify when the postoperative SBP was measured (line 86)
Answer: Thank for your very kindly advice. We sincerely revised our study like below.
Postoperative SBP was measured the first three times (postoperative 1 hour, 2 hours and 3 hours) in the postoperative ward and set as its average.
4) Please clarify when the postoperative blood tests were taken (line 94)
Answer: Thank for your very kindly advice. We sincerely revised our study like below.
….difference from baseline in pre- and post-operative platelet count (base-line: 150,000 /ul), PT INR (baseline: 1.0) and aPTT (baseline: 35 seconds), spinal disease, surgical level range, intraoperative time, anti-hypertensive drug. Postoperative blood tests were taken in the postoperative 3 hours.
5) Table 1 is a bit confusing - is it the n number and then the s.d. in brackets? Seems to apply to the top few rows but not for INR, which seems to be average INR and (range)?
Answer: Thank for your very kindly advice. We sincerely revised our study like below.
Number =1296
Continuous data were presented as median and interquartile ranges (IQRs).
Preoperative platelet difference (10,000/㎕) (median, IQR)
Preoperative INR difference (median, IQR)
6) Please describe the effect of anticoagulant use in line 173 and your institute’s policy in stopping these pre-operatively
Answer: Thank for your very kindly advice. We sincerely revised our study like below.
On the other hand, anticoagulant or anti-platelet drugs were not related with incidence of PSEH in our study. Because, for reducing of perioperative bleeding tendency, anticoagulant or anti-platelet drugs were withdrawn during at least five days before surgery in our hospital.
Minor points
1) Please add the OR and p-value for postoperative SBP in the abstract (“Increase of postoperative SBP had statistically significant effect on 29 the incidence of symptomatic PSEH…”)
Answer: Thank for your very kindly advice. We sincerely revised our study like below.
Increase of postoperative SBP (OR 1.68, p = 0.048) had statistically significant effect on the incidence of symptomatic PSEH.
2) Please note that INR is not always the best measure for monitoring the effect of the novel anticoagulants – another limitation
Answer: Thank for your very kindly advice. We sincerely revised our study like below.
This study has some limitations. The main limitation of our study is that data were retrospectively analyzed. For example, INR is not always the best measure for monitoring the effect of the novel anticoagulants. Therefore, retrospective INR data could be not accuracy. Another limitation is that postoperative lumbar MRI could not examined in all patients. Therefore, we could not analyzed the amount and location of PSEH, severity of neural compression, and asymptomatic PSEH by MRI imaging.

Reviewer 3 Report
In this article, the Authors investigate the role of some of the most common risk factors in the development of PSEH. The paper is well structured although it is not innovative and although it is retrospective, the results are also clearly presented.
In the end, their results only confirm what every surgeon has always suspected, namely, that these types of hematomas are influenced by sudden changes in blood pressure, platelet deficits, or excessive blood loss while, on the contrary, other classic risk factors such as diabetes or BMI are not so relevant.
I believe that although the design and results are not so revolutionary the work is still well written and can be published in this journal.
Author Response
Reviewer 3
Comments and Suggestions for Authors
In this article, the Authors investigate the role of some of the most common risk factors in the development of PSEH. The paper is well structured although it is not innovative and although it is retrospective, the results are also clearly presented.
In the end, their results only confirm what every surgeon has always suspected, namely, that these types of hematomas are influenced by sudden changes in blood pressure, platelet deficits, or excessive blood loss while, on the contrary, other classic risk factors such as diabetes or BMI are not so relevant.
I believe that although the design and results are not so revolutionary the work is still well written and can be published in this journal.
Answer: Thank for your very kindly Comments.
